# Gene Cloning, Tissue Expression Profiles and Antiviral Activities of Interferon-β from Two Chinese Miniature Pig Breeds

**DOI:** 10.3390/vetsci9040190

**Published:** 2022-04-15

**Authors:** Aziz Ullah Noor, Zhanyu Du, Chengyi Song, Huipeng Lu, Xiaohui Zhou, Xiaoming Liu, Xinyu Zhang, Huaichang Sun

**Affiliations:** 1The College of Veterinary Medicine, Jiangsu Co-Innovation Center for Prevention and Control of Important Animal Diseases and Zoonoses, Yangzhou University, Yangzhou 225009, China; drshinwari72@gmail.com (A.U.N.); luhuipeng66@126.com (H.L.); Zxh160096@yeah.net (X.Z.); L997758335@163.com (X.L.); zxy@yzu.edu.cn (X.Z.); 2The College of Animal Science and Technology, Yangzhou University, Yangzhou 225009, China; yuzhan.du@163.com (Z.D.); cysong@yzu.edu.cn (C.S.)

**Keywords:** Chinese miniature pigs, interferon-β gene cloning, tissue expression profile, antiviral activities

## Abstract

The porcine interferon (PoIFN) complex represents an ideal model for studying IFN evolution which has resulted from viral pressure during domestication. Bama and Banna miniature pigs are the two Chinese miniature pig breeds that have been developed as laboratory animal models for studying virus infection, pathogenesis, and vaccine evaluation. However, the PoIFN complex of such miniature pig breeds remains to be studied. In the present study, we cloned PoIFN-β genes from Bama and Banna miniature pigs, detected their PoIFN-β tissue expression profiles, prepared recombinant PoIFN-β (rPoIFN-β) using the *E. coli* expression system, and measured their antiviral activities against three different pig viruses. At the amino acid sequence level, PoIFN-βs of the two miniature pig breeds were identical, which shared 100% identity with that of Congjiang Xiang pigs, 99.4–100% identity with that of domestic pigs, and 99.5% identity with that of three species of African wild boars. The tissue expression profiles of PoIFN-β mRNA differed not only between the two miniature pig breeds but between miniature pigs and domestic pigs as well. The four promoter domains of PoIFN-β of the two miniature pig breeds were identical with that of humans, domestic pigs, and three species of African wild boars. The recombinant PoIFN-β prepared from the two miniature pig breeds showed dose-dependent pre-infection and post-infection antiviral activities against vesicular stomatitis virus, porcine respiratory and reproductive syndrome virus, and pig pseudorabies virus. This study provided evidence for the high sequence conservation of PoIFN-β genes within the *Suidae* family with different tissue expression profiles and antiviral activities.

## 1. Introduction

The porcine interferon (PoIFN) complex represents an ideal model for studying IFN evolution that resulted from viral pressure during domestication [1]. Type I PoIFNs comprise at least 39 functional genes, which are classified into 17 IFN-α subtypes, 11 IFN-δ subtypes, and 7 IFN-ω subtypes, as well as a single-type of IFN-β, IFN-ε, IFN-κ and IFN αω [2]. Among these IFN subtypes, interferon-beta (IFN-β) is considered to be the “primary” type I IFN due to its broad expression in most nucleated cells, including macrophages, B cells, T cells, NK cells, osteoblasts, and endothelial cells [3]. When binding to type I IFN receptor (IFNAR1 and IFNAR2), type I IFNs signal through classical JAK/STAT pathway, leading to stimulation of many genes with antiviral, immunomodulatory and antiproliferative activities on different cell types. Therefore, recombinant IFNs have been widely used to treat viral diseases. For example, IFN-β has been used to treat multiple sclerosis [4]. IFN-α has been used to treat types B and C hepatitis [5]. Feline recombinant IFN-ω has been certified to treat viral infections such as feline leukemia virus, feline immunodeficiency and parvovirus [6]. Although its gene sequence is available in GenBank, PoIFN-β remains to be well characterized, in miniature pigs in particular.

Unlike domestic pigs which are difficult to handle and expensive to use, miniature pigs are easy to handle and convenient for taking repeated samples, and thus represent ideal models for studying porcine and human diseases, as well as vaccine and drug development [7]. Chinese Bama miniature pig breed is one of the few miniature pig breeds in the world [8,9], which was inbred from Bama Xiang pigs by long-term selective inbreeding [10]. The Banna miniature pig breed was inbred by the half-siblings of Diannan small-ear pigs [11,12]. Both Banna and Bama miniature pigs have been developed as laboratory animal models for studying virus infection, pathogenesis, and vaccine evaluation [11,13]. Unlike domestic pigs which have been domesticated for 10,000 years [14], miniature pigs have been less domesticated and are exposed to different virus pressures [2,15]. These miniature pig breeds are genetically stable and disease-resistant [16,17], but their IFN complexes remain to be characterized. The objectives of the present study were as follows: (1) clone PoIFN-β genes; (2) detect tissue expression profiles; (3) prepare recombinant PoIFN-β; and (4) characterize antiviral activities of the two Chinese miniature pig breeds.

## 2. Materials and Methods

### 2.1. Cells and Viruses

Marc-145 cells, PK-15 cells and MDBK cells (ATCC, Rockville, MD, USA) were cultured in Dulbecco’s modified Eagle medium (DMEM, Gibeco, Shanghai, China) supplemented with 10% fetal bovine serum (FBS), streptomycin (100 μg/mL) and penicillin (100 U/mL). Porcine reproductive and respiratory syndrome virus (PRRSV) VR2332 strain (ATCC, Rockville, MD, USA) was cultured and titrated on Marc-145 cells as previously described [18]. Vesicular stomatitis virus (VSV) New Jersey strain (ATCC, Rockville, MD, USA) was cultured and titrated on MDBK cells [19]. Porcine pseudorabies virus (PRV) Bartha K61 strain (ATCC, Rockville, MD, USA) was cultured and titrated on PK-15 cells [20]. All viruses were titrated as 50% tissue culture infective dose (TCID_50_).

### 2.2. Animal and Ethical Approval

Three six-month-old healthy Banna and Bama miniature pigs were brought from the Beijing Institute of Zoology, Chinese Academy of Sciences, and kept at the Experimental Pig Farm of Yangzhou University. The experiment was carried out according to the recommendations in the Guide for the Care and Use of Laboratory Animals of Yangzhou University. The protocol was approved by the Medical Experimental Animal Center of Jiangsu Province (Permit Number: 2140880).

### 2.3. Tissue Sampling

Anesthesia was executed as previously described [21]. Intramuscular injection of ketamine (10 mg/kg) and atropine sulfate (1 mg) was administered as the pre-anesthesia, followed by injection with 1% thiopental (0.5 mL/kg) to achieve complete anesthesia. The experimental animals were sacrificed by femoral artery bleeding. Different tissues were collected from the sacrificed animals, including brain, heart, lung, spleen, liver, lymph node, skin, intestine kidney, and uterus. Each tissue was cut into 1-g pieces, kept into cryotubes, and preserved in liquid nitrogen until further use.

### 2.4. Gene Cloning and Sequence Analysis

Genomic DNA was extracted from the spleen tissues of Banna and Bama miniature pigs using Genomic DNA Extraction Kit (Tiangen Biotech, Beijing, China). The coding regions of PoIFN-βs were amplified from the genomic DNA using High-Fidelity PCR Taq System (TaKaRa, Beijing, China) with primers listed in Table 1. The amplicons were cloned into pMD19-T TA cloning vector (TaKaRa, Beijing, China) and transformed into DH5α *E. coli*. The amplicon-containing vectors were prepared and sequenced in two directions using vector sequencing primers (TaKaRa). The obtained sequences were assembled using the SeqMan program (DNASTAR Lasergene, Madison, WI, USA), and the consensus open reading frames (ORFs) were translated into the amino acid sequences using the EditSeq program. The translated sequences were blasted against PoIFN sequences in GenBank for homology, conserved cysteine residues and potential N-glycosylation sites using the BLAST algorithm (https://blast.ncbi.nlm.nih.gov/Blast.cgi, accessed on 2 December 2022). The amino acid sequences were further aligned with that of 11 different animal species (Appendix A) for sequence homologies (Clustal W scores) using the MegAlign program [22].

### 2.5. Sequence Analysis of PoIFN-β Regulatory Elements

The genomic DNA was extracted from the spleen tissues of Bama and Banna minipigs using a Genomic DNA Extraction Kit. The regulatory elements of PoIFN-β genes were amplified by High-Fidelity PCR Taq System using the primers listed in Table 1. After pre-denaturation for 5 min at 95 °C, 5-cycle touch-down PCR (94 °C, 20 s; 60–56 °C, 20 s; 72 °C, 20 s) and then 30-cycle PCR (94 °C, 20 s; 55 °C, 20 s; 72 °C, 20 s) were performed, followed by final extension for 10 min at 72 °C [23]. The PCR products were cloned into pMD19-T TA cloning vector and sequenced in two directions using vector sequencing primers as described. The obtained sequences were aligned for the four promoter domains (PRDs) in PoIFN-β genes with that of humans, domestic pigs, other miniature pig breeds and three species of African wild boars.

### 2.6. Real-Time Quantitative RT-PCR

Tissue expression of PoIFN-β mRNA in different tissues of Banna and Bama miniature pigs was analyzed by Real-Time SYBR Green RT-PCR System (LightCycler ^R^ Nano (Roche)) as previously described [2], with validated primers listed in Table 1. Total RNA was extracted from different tissues with MiniBEST Universal RNA Extraction Kit (TaKaRa). Each reaction involved pre-denaturation for 20 s at 95 °C, followed by 40 cycles of 95 °C for 5 s, 64 °C for 20. The melting curve analysis was performed to check the specific product. Each sample was analyzed in triplicates with no template control (NTC). Relative gene expression data in different tissues were normalized against critical threshold (Ct) values of the house-keeping gene (GAPDH). The relative expression index (^2-^^△△^Ct) was determined in comparison to the average expression levels of control samples [24].

### 2.7. Expression and Purification of rPoIFN-β

The coding sequence for the mature peptide of PoIFN-β was amplified from the sequencing vector using High Fidelity Taq DNA Polymerase (TaKaRa) and gene-specific primers listed in Table 1. The PCR product was cloned into pET-30a (+) vector (Novagen, Madison, WI, USA) by NdeI and XhoI digestion. The expression of His-tagged rPoIFN-β was under the control of T7 promoter. After transformation into BL21 (DE3) E. coli, a single colony was cultured overnight in 5 ml of Luria-Bertani (LB) medium containing kanamycin (50 μg/mL). The overnight culture was 1:100 diluted with 10 mL of 2 × YT medium (10 g yeast extract, 16 g tryptone and 5 g NaCl) containing the same antibiotic. After growth for 4 h at 37 °C, the expression of recombinant protein was induced with 0.2 mM Isopropyl β-d-1- thiogalactopyranoside (IPTG) for 6 h at 37 °C. After 10-min centrifugation at 6000× *g*, the pellet was suspended in 1 mL of lysis buffer (50 mM Tris-HCl, 50 mM NaCl, 5% glycerol, pH7.2). After sonication treatment (40 w, 10 s, 20 s intervals, 5 min) and 10-min centrifugation at 12,000× *g* at 4 °C, the supernatant was collected and rPoIFN-β was purified under natural condition by Ni-NTA Metal Affinity Chromatography (CWBIO, China). Before and after purification, the protein samples were analyzed for purity by 12% SDS-PAGE. The purified PoIFN-β protein was quantified by Bicinchoninic acid assay (BCA.) (Sangon Biotech, Shanghai, China). Since the presence of His-tag at the C-terminus, the rPoIFN-β was identified by Western blotting using mouse anti-His-tag mAb (1:3000, Sangon Biotech) as the first antibody, and HRP-conjugated goat anti-mouse IgG (1:10,000; Sangon Biotech) as the second antibody. The hybridization signals were developed with a High-Sensitive ECL Chemiluminescence Detection Kit (Vazyme, Nanjing, China) and scanned on Gel Documentation System (Tanon, Shanghai, China).

### 2.8. Antiviral Assay

The antiviral activities of rPoIFN-β against PRV, VSV, and PRRSV were measured on PK-15, MDBK or Marc-145 cells using cytopathic effect (CPE) inhibition assay as previously described [2,15]. To detect the pre-infection antiviral activities of rPoIFN-β, the cells were seeded on 96-well plates and grown for 24 h at 37 °C in 5% CO_2_. The rPoIFN-β was 10-fold serially diluted and added in 8 duplicates to wells. After incubation for 24 h at 37 °C, the cells were infected with VSV (100 TCID_50_), PRV (0.2 TCID_50_) or PRRSV (0.1 TCID_50_), with untreated and uninfected cells as the normal control, and untreated and virally infected cells as the virus control. To detect the post-infection antiviral activities of rPoIFN-β, the cells were infected with VSV, PRV or PRRSV as described, and treated in triplicates with rPoIFN-β (20 ng/mL) at 2, 4, 8, 16 or 24 h post infection. The cells were stained with 1% crystal violet in 15% ethanol, and extracted with 70% ethanol and 1% acetic acid. The OD_580_ values were measured on Multifunctional ELISA Reader (BioTek, Winooski, VT, USA). The percentage of CPE protection was calculated with the formula (Vt − Vi)/(Vt − V0) × 100, where Vt, Vi, and V0 indicate the highest occurrence of viral infection in mock-treated cells, the averaged OD_580_ value of IFN-treated cells and the averaged OD_580_ value of cells without adding virus and IFN, respectively.

### 2.9. Statistical Analysis

The GraphPad Prism version (9.0.1) was used for Graphical illustrations. Both *t*-test and one-way ANOVA were used to evaluate the significance of data difference by following Tukey’s Post Hoc Multiple Comparison Test. *p* < 0.001 were considered as statistical significance.

## 3. Results

### 3.1. Cloning and Sequence Analysis of PoIFN-β Genes

By using the gene-specific primer pair, the expressed 561-bp products were amplified from the spleen tissues from Bama and Banna miniature pigs (Figure 1). Sequence analysis showed that the two PoIFN-β genes were identical, which shared 97.9–100% homology at the nucleotide sequence level, or 98.4–100% homology at the amino acid sequence level, with that of Congjiang Xiang miniature pig, African miniature pig, domestic pigs and three African wild boars (Table 2). The 561-bp ORFs encoded polypeptides of 186 amino acids (aa), with a predicted 21-aa signal peptide, three conserved cysteine residues at positions 17, 31 and 140, and a putative *N*-glycosylation site (Asn/Asp-X-Ser/Thr) (2) at position 80 of the mature peptide (Figure 2). Further sequence alignment showed amino acid substitutions, signal peptides, conserved cysteine residues and *N*-glycosylation sites of IFN-βs of the two Chinese miniature pig breeds as compared with that of humans, domestic pigs and other seven animal species (Figure 2).

### 3.2. Sequence Analysis of PoIFN-β Regulatory Elements

By using the primer pair specific for PoIFN-β promoter, two expected 259-bp products were amplified from the genomic DNA from Bama and Banna miniature pigs. Sequence analysis showed that the four PoIFN-β promoter domains (PRDs) of the two miniature pig breeds were identical to that of humans, domestic pigs and three African wild boars, including interferon response factors IRF3 and IRF7 in PRDI and PRDIII, NF-kB in PRDII, and ATF and Jun modules in PRDIV (Figure 3).

### 3.3. Detection of PoIFN-β Tissue Expression Profiles

Real-time quantitative RT-PCR analysis showed that, among 10 different tissues detected, the tissue expression profiles of PoIFN-β were significantly different between the two miniature pig breeds (Table 3). For example, low levels of PoIFN-β mRNA were detected in heart, liver and brain of Bama miniature pigs, but not in that of Banna miniature pigs. In addition, lung was the highest PoIFN-β expression tissue of Bama miniature pigs with a relative expression index of 1.398, while liver was the highest PoIFN-β expression tissue of Banna miniature pigs with a relative expression index of 1.784.

### 3.4. Expression and Identification rPoIFN-β

After induction with IPTG, SDS-PAGE analysis showed an expected 21-kDa extra protein was detected in pET30a-PoIFN-β transformed *E. coli*, but not in the empty pET30a transformant and un-induced pET30a-PoIFN-β transformant (Figure 4). The rPoIFN-β was expressed as a soluble protein which was present in the supernatant of centrifuged bacterial lysate. The recombinant protein was purified to single band by Ni-NTA affinity chromatography (Figure 4). Western blotting analysis showed that the purified protein was recognized by mAb against His-tag which was present at the C-terminus of rPoIFN-β (Appendix A).

### 3.5. Detection of Antiviral Activities of rPoIFN-β

To detect the pre-infection antiviral activities of rPoIFN-β, MDBK, PK-15 or Marc-145 cells were treated with different concentrations of the recombinant protein before VSV, PRV or PRRSV infection. CPE inhibition assay showed that the rPoIFN-β had dose-dependent pre-infection antiviral activities against three pig viruses (Figure 5). Compared to the untreated virus control, rPoIFN-β showed significant anti-VSV and anti-PRRSV activities at doses of ≥0.02 ng/mL or anti-PRV activity at doses of ≥0.2 ng/mL. Among the three pig viruses tested, VSV was the most sensitive virus to rPoIFN-β (91% protection at 200 ng/mL), followed by PRRSV (79% protection at 200 ng/mL) and PRV (70% protection at 200 ng/mL).

To detect the post-infection antiviral activities of rPoIFN-β, MDBK, PK-15 or Marc-145 cells were infected with VSV, PRV or PRRSV, and then treated with a fixed concentration (20 ng/mL) of rPoIFN-β at different time points post-infection. CPE inhibition assay showed that rPoIFN-β exhibited post-infection antiviral activities against the three pig viruses, which declined as the time extension of virus infection (Figure 6). At 2 h post-infection, treatment with rPoIFN-β showed 91% CPE protection in VSV-infected cells, 78% CPE protection in PRV-infected cells or 74% CPE protection in PRRV-infected cells, which declined to 45%, 11%, and 7% protection by 24 h post infection, respectively.

## 4. Discussion

Porcine IFN-complex represents an ideal model for studying IFN evolution resulted from viral pressure during domestication, and thus subtype-diversification of type I IFNs deserves extensive antiviral analyses [1]. PoIFN-β genes have been cloned from domestic pigs, Congjiang Xiang pigs, Guizhou Baixiang pigs, African miniature pigs and three species of African wild boars, but not from Bama and Banna miniature pigs. In addition, the molecular structures, tissue expression profiles, and antiviral activities of PoIFN-β remain to be characterized. In the present study, we cloned PoIFN-β genes from the two Chinese miniature pig breeds, which have been developed as laboratory animal models for studying human and pig diseases [11,13]. Unexpectedly, PoIFN-β genes of the two miniature pig breeds were identical, which shared 98–100% homology with that of domestic pigs. Surprisingly, PoIFN-β genes of the two miniature pig breeds shared more than 99% homology with that of three species of African wild boars, which belong to different Genus or subfamily within the *Suidae* family [25]. Further sequence alignment showed that IFN-β genes of 10 animal species encoded 186-aa polypeptides, which were a single amino acid shorter than that of human IFN-β. Nevertheless, all of IFN-β proteins had 21-aa signal peptides, three conserved cysteine residues that are important for disulfide bond formation and structure stabilization [26,27], and potential *N*-glycosylation sites at position 80 of the mature peptides. These data indicate the high conservation of PoIFN-β, not only within the *Suidae* family, but between different animal species as well.

Although sharing high sequence similarity in coding ORFs, the promoter regions of type I PoIFNs are quite different. This suggests that the expression of type I PoIFNs is different in respect to tissue/cell types and/or stimuli [2]. It has been shown that PoIFN-β is expressed in all of seven tissues from domestic pigs, with the highest expression level in the skin [2]. In the two Chinese miniature pig breeds, however, the expression of PoIFN-β in their skin tissues was not detectable. In addition, the tissue expression profiles of PoIFN-β between the two miniature pig breeds were also different. To explain the differential expression, the core regions of PoIFN-β promoters were cloned from the two miniature pig breeds. Sequence alignment showed that the four PoIFN-β promoter domains of the two miniature pig breeds were identical to that of domestic pigs, humans and three species of African wild boars. These data suggest that the differential expression of PoIFN-β may be caused by different environmental stimulators, not by promoter sequence variation.

It has been shown that the rPoIFN-β expressed in eukaryotic cells has strong CPE protection in PRRSV-infected macrophages and PRV-infected PK-15 cells, but not in the virally infected Marc-145 cells [2]. In this study, the rPoIFN-β gene from Bama miniature pigs was expressed in *E. coli* as a His-tagged protein, and purified to a single band by affinity chromatography without overt cytotoxicity (data not shown). CPE inhibition assay showed that the rPoIFN-β had dose-dependent pre-infection antiviral activities against all of three pig viruses tested. Unexpectedly, the rPoIFN-β showed strong CPE protection in all of three virally infected cell types. Although the detailed reason remains to be defined, the discrepancy in antiviral activities between the two rPoIFN-β proteins may be caused by different doses used for the antiviral assay. The post-infection antiviral activities of IFNs suggest their potential therapeutic effects against virus infections [28,29,30]. In this study, the rPoIFN-β prepared from the Chinese miniature pigs showed significant post-infection antiviral activities against all of three pig viruses tested. This suggests that the rPoIFN-β prepared in this study could be used to treat pig viral diseases.

## 5. Conclusions

This study provided the evidence for high conservation of PoIFN-β within the *Suidae* family with differential tissue expression profiles between two Chinese miniature pig breeds and domestic pigs, which was not due to the sequence variation in PoIFN-β promoters. The rPoIFN-β prepared in vitro had strong antiviral activities against three pig viruses. However, more in vivo tests will be needed to conclude the rPoIFN-β can be used to treat animals, such as maximum tolerated dose to treat swine viral diseases.

## Figures and Tables

**Figure 1 vetsci-09-00190-f001:**
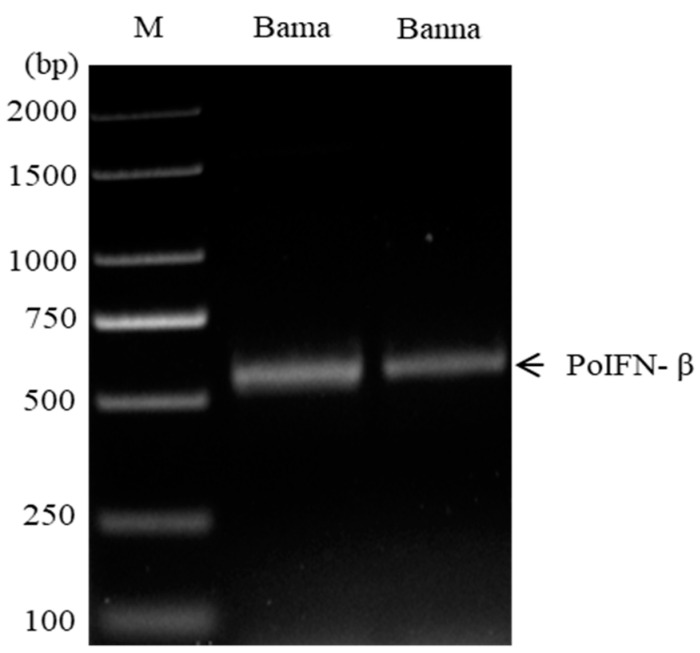
Amplification of PoIFN-β genes from two Chinese miniature pig breeds.

**Figure 2 vetsci-09-00190-f002:**
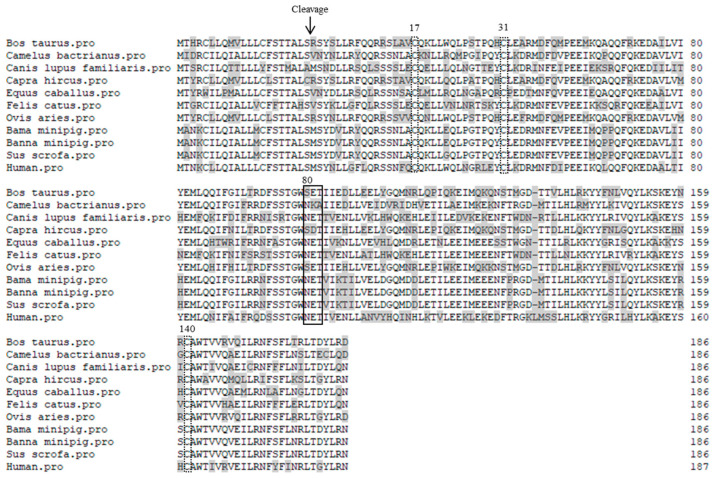
Amino acid sequence alignment of IFN-β proteins between different animal species. The cleavage site of signal peptides is indicated with an arrow. Amino acid substitutions are shadowed. The conserved cysteine residues at the indicated positions are dot line-boxed. Putative *N*-glycosylation sites are solid line-boxed.

**Figure 3 vetsci-09-00190-f003:**
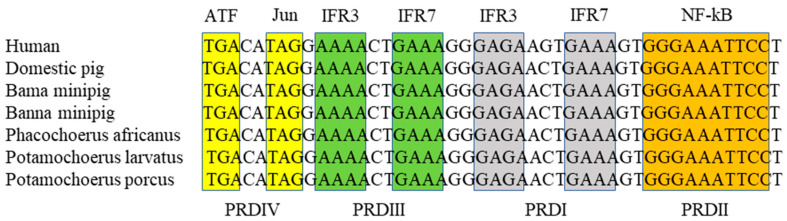
Sequence alignment of the promoter domains in PoIFN-β of different pig species. The promoter domains (PRDs) in PoIFN-β of two Chinese miniature pig breeds were identical with that of humans, domestic pigs, and African wild boars.

**Figure 4 vetsci-09-00190-f004:**
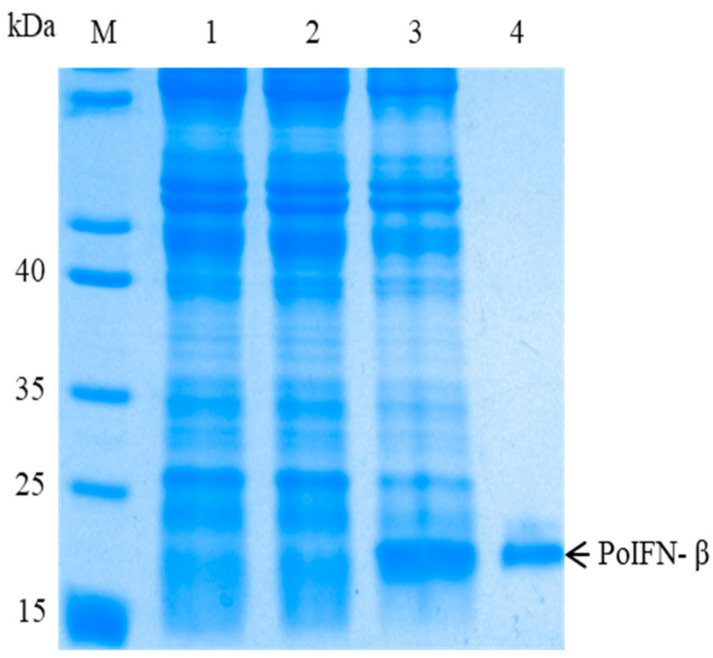
Expression and purification of rPoIFN-β from Bama miniature pig. The protein samples were run on 12% SDS-PAGE. 1–4 indicate pET-30a-tansformed *E. coli*, pET30a-PoIFN-β transformed *E. coli* before IPTG induction, pET30a-PoIFN-β transformed *E. coli* after IPTG induction and purified rPoIFN-β, respectively.

**Figure 5 vetsci-09-00190-f005:**
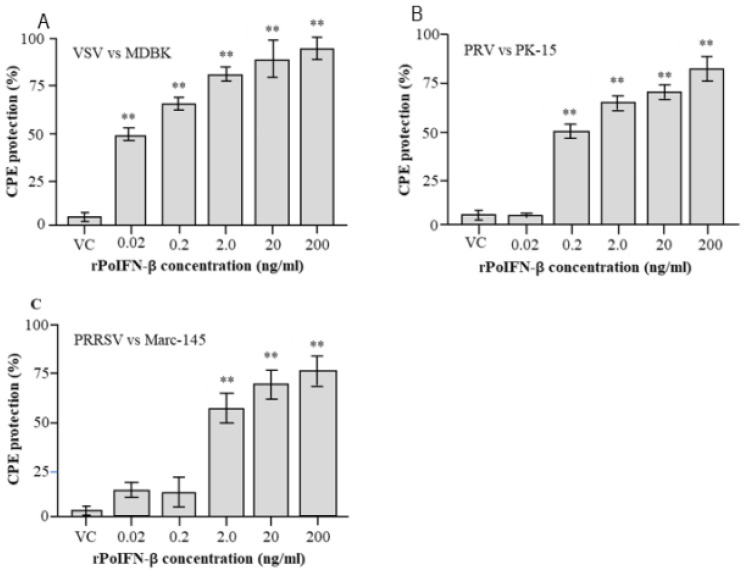
Detection of pre-infection antiviral activities of rPoIFN-β. Three cell lines (**A**–**C**) were treated with the indicated concentrations of rPoIFN-β for 24 h and then infected with the three pig viruses. The pre-infection antiviral activities were measured in triplicates by CPE inhibition assay. ** indicates significant difference (*p* < 0.01) in CPE protection of rPoIFN-β as compared with the virus control (VC).

**Figure 6 vetsci-09-00190-f006:**
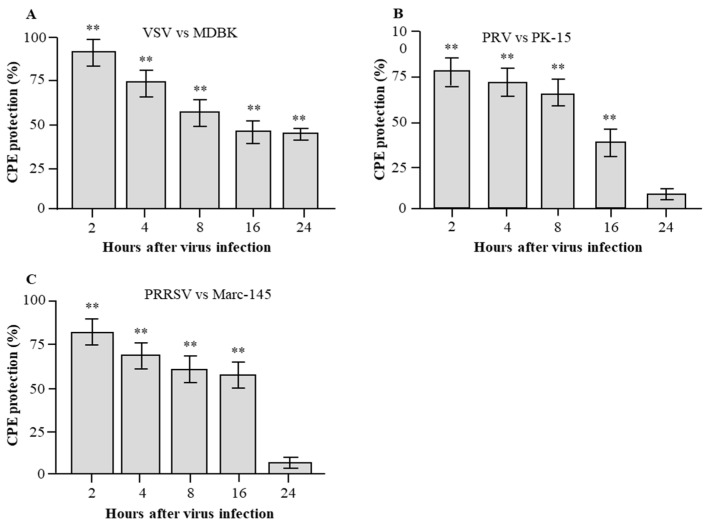
Detection of post-infection antiviral activities of rPoIFN-β. Three cell lines MDBK cells (**A**), PK-15 cells (**B**) and Marc-145 cells (**C**) were infected with the indicated viruses VSV (**A**), PRV (**B**) or PRRSV (**C**) and then treated with a fixed concentration of rPoIFN-β. At different time points post infection, the post-infection antiviral activities were measured in triplicates by CPE inhibition assay. ** indicates significant difference (*p* < 0.01) in CPE protection of rPoIFN-β as compared with the virus control (VC).

**Table 1 vetsci-09-00190-t001:** PCR primers used in this study.

Gene		Primer (5′-3′)	Amplicon	Reference
PoIFN-β ^a^	Sense	ATGGCTAACAAGTGCATCC	561	This study
	Anti-sense	TCAGTTCCGGAGGTAATCT		
PoIFN-β ^b^	Sense	AAATCGCTCTCCTGATGTGTT	539	This study
	Anti-sense	TCAGTGGTGGTGGTGGTGGTGGTTCCGGAGGTAATCTGTAA		
PoIFN-β ^c^	Sense	ATGTCAGAAGCTCCTGGGACAGTT	246	Sang et al.
	Anti-sense	AGGTCATCCATCTGCCCATCAAGT		
PoIFN-β ^R^	Sense	AGTGTTGGATGAATGCTAACAA	259	This study
	Anti-sense	TGGTGGAGAAACACATCAGG		
GAPDH	Sense	TGGYATCGTGGAAGGRCTCAT	370	Sang et al.
	Anti-sense	RTGGGWGTYGCTGTTGAAGTC		

^a^ Primer pair for gene cloning; ^b^ Primer pair for expression vector construction; ^c^ Primer pair for detection of differential expression by real-time quantitative RT-PCR; ^R^ Primer pair for cloning of regulatory element of PoIFN-β gene.

**Table 2 vetsci-09-00190-t002:** Sequence homologies (Clustal W scores) of PoIFN-βs within *Suidae* family.

Species	Homology (Clustal W Score)		
	ORF%	Protein%	GenBank Accession No
Bama minipig	Reference	Reference	OL446997
Banna minipig	100	100	OL446998
Congjiang Xiang minipig	100	100	MH538100
Guizhou Baixiang minipig	99.0	98.4	JF906509
African minipig	99.0	99.5	JN391525
Domestic pig	100	99.5	GQ415073
	99.8	100	NM_0010039231
	98.8	97.9	KF4147411
	99.8	99.4	EU744562
	99.8	99.5	M86762
Phacochoerus africanus	99.0	99.5	JN391529
Potamochoerus porcus	99.0	99.5	JN391527
Potamochoerus larvatus	99.0	99.5	JN391526

**Table 3 vetsci-09-00190-t003:** Tissue expression profiles of PoIFN-β of two Chinese miniature pig breeds.

Minipig	Skin	Uterus	Heart	Lung	Intestine	Liver	Spleen	Lymph Node	Kidney	Brain
Banna	-	0.375	-	1.142	0.488	1.784	0.551	0.474	-	-
Bama	-	0.044	0.204	1.398	0.465	0.329	0.135	0.008	0.239	0.041

## Data Availability

The coding sequences for PoIFN-β of the two Chinese miniature pig breeds are available in GenBank (Accession numbers: OL446997 and OL446998). The expression vectors for rPoIFN-β preparation can be obtained from the corresponding author if requested.

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
