# Peer review of "Gene Cloning, Tissue Expression Profiles and Antiviral Activities of Interferon-β from Two Chinese Miniature Pig Breeds"

_vetsci, 2022, doi:10.3390/vetsci9040190_

Round 1
Reviewer 1 Report
The authors purported to describe PoINF-β regulation and heterologous expression efficacy from two miniature Chinese pig breeds in comparisons to orthologs.
These are minor corrections:
Line 38: Interferon-beta (IFNβ) instead of IFNβ only
Line 181: there is one round bracket too much
Line 215: a reference seems to be missing and probably replacing “(36)”
Lines 230-231: Please rewrite the sentences: The mature protein of 186...IFab domain.
Line 390 is probably “address” instead of “dress”.
You need definitively to elaborate more on these corrections:
Line 203-206: This is not really a “touchdown PCR” that you performed. The reference is also missing.
Table 1. Please correct typo in References and directly indicate the start and end of individual primers with 5’ and 3’ respectively. Table 2 was fused with Table 1: the Tables should be separated.
In Table 2: there is also a problem with the sequence ID, you would need to list the accession numbers for the sequences as indicated under 3.2 of the Results. Additionally, please describe the nucleotide and aa homology also for each domestic pig separately. These data are missing in Table 2.
Under the title of 3.2 of the Results, you also indicated Figure legend descriptions: this is unnecessary. Please also do not make up a new definition for orthologs (line 256). Instead stick to the accepted definition: “A homologous gene that is related to those in different organisms by descent from the DNA of a common ancestor and that may or may not have the same function.”
What is the meaning of “6” in Figure 2 under the aa- alignment?
Why did you not take NP_002167.1 for the human IFN-b comparison in your alignment in Figure 2? It is full length and has therefore the same length as the PoIFN-β.
In Figure 3b. the Western blot: something is wrong with the molecular marker (kd) indication. Can you also comment on the double band in lane 4 of the SDS-PAGE Gel? Why is it not seen in lane 2 of the Western blot analysis.
Lines 316-317 should indicate (Figure 5a and 5c)… (Figure 5a).
Figure 6 legends: Initially you described the PRD and suddenly you also include the PolFN-β enhancer. Promotors and enhancers are different transcriptional elements.
To table Table 3. I must assume that you have analyzed the RNA expression in the different tissues. Nevertheless you did not describe the RT reaction under 2.12 of the Materials and Methods. It would have been better to use “run-ons” to answer this question. The chosen experiments are a quick and dirty way to access your question.
Please adjust discussion accordingly and keep to formatting rules “4.” Cursive.
Animal and ethical approval: This is definitively inadequately written. What about the approval from outside the university or an authorization number to appropriately kill these animals and removing their organs.
Author Response
Dear Editor,
We appreciate and accept the comments and suggestions from the two reviewers. The manuscript has been systemically revised as suggested. The major revisions include reorganization of manuscript, English polishing, adding animal and ethical approval, deletion of non-relevant references and addition of relevant references. Due to the comprehensive correction, the grammar errors and wrong sentences in the previous manuscript have been corrected or avoided.
Thanks for your kind suggestions and we look forward to hearing from you very soon

Reviewer 2 Report
General comments:
I think the work presented in the paper is relevant for the field and brings novelty. However, the manuscript is poorly written and needs to be revised by a language expert. There is a significant number of confusing sentences and misspelling that makes it difficult to understand what the authors are trying to communicate.
References were presented in several different formats. Citation and references should be formatted according to the journal guidelines.
Specific comments:
Page 3: Material and methods part “2.7. Prokaryotic Expression of rPoIFN-β” needs to be better detailed. What equipment was used to measure the OD value? What was the interval of time between the measurement? The methods description from Line 135 to 140 is confusing and should be re-written to make the information clear. The authors should fully describe each step.
Line 133: “37°C hours at 30°C,” needs to be corrected.
Line 151-152: “The incubation of the membrane was repeated for 3 hours at room temperature with mouse anti-His serum” sentence is confusing. Was the membrane incubated with a blocking solution for a second time? Mouse anti-His serum, should be replaced for mouse anti-His antibody.
Page 4: Material and methods part “2.9. Dose-dependent antiviral profile of rPoIFN-β” needs to be re-written and better detailed to make the information clear. How many concentrations were tested (Line 168)? How were the treatments distributed in the plate?
Line 187: What cells were used to seed the wells?
Line 192-193: should be re-write to make the information clear.
Line 206: “Final cycle” should be substituted for Final extension step.
Line 225: “and DNA sequencing. (Figure 1B). The analysis” should be added the description of the results of Figure 1B.
Page 5-6: Table 2 should be separated from Table 1.
Line 285: “The 97% purified 21kDa” The authors should add more details to the sentence to make the information clearer. Is it referencing to 21kDa proteins?
Page 9: Results part “3.4. Cytotoxicity of the PoIFN-β” should include the results of the viability assay. It will improve the manuscript and make the conclusion more solid.
Line 316-317: “(Figure 3B and C)” and “(Figure 3(a))” This is actually figure 5, the authors should change to Figure 5B and C and Figure 5A.
Page 11-12: Results part “3.8. Differential expression of rPoIFN-β different tissue” should be re-write to make the information clear. How was the index calculated?
Line 394-399: This conclusion is too strong and the data does not support it; The sentence should be re-written to clarify that it is the promoter domain region that is identical between the species, but not the IFNβ gene.
Figure 1 legend: “The PoIFN- β sequence was cloned into the 239 vectors at the NdeI and Xhol digestion sites. pET30a-PoIFN-β plasmid consisted of a His tag and the mature PoIFN- β 240 sequence.” should be moved to the beginning of the legend description, before “(a)”.
Figure 3b: Why the rPoIFN-b of 1b has a different size of 1a? Shouldn't both be a 21kDa protein?
Figure 4 legend: it should specify which experiments were analyzed in 5 or 8 duplicates.
Figure 5: Include untreated and virus control data to the graphs. It will improve the understanding of the data.
Author contributions section: “ZD carried out animal raising and blood collection”. However, the paper methodology and results section do not describe the use of blood samples in this paper. It should be corrected.
Data availability statement: The GenBank access numbers given by the authors did not work when typed in the website: https://www.ncbi.nlm.nih.gov/genbank/ . It should be corrected.
Author Response

(The authors gave the same response as above.)

Round 2
Reviewer 1 Report
Under Materials and Methods, in section 2.7: you would need to indicate how did you measure the recombinant protein concentrations.
"GenBank Accession" in Table 2 should be complemented with No., thus it should read GenBank Accession No.
Author Response
Dear Editor,
We appreciate and accept the comments and suggestions from the two reviewers.
- Under Materials and Methods, in section 2.7: you would need to indicate how did you measure the recombinant protein concentrations.
Reponse1. The recombinant protein concentrations measure method is added in section 2.7.
2 "GenBank Accession" in Table 2 should be complemented with No., thus it should read GenBank Accession No.
Reponse2. In Table 2 word "No" is added as suggested.

Reviewer 2 Report
General comments:
I think the work presented in the paper is relevant for the field and brings novelty. The authors did almost all the requested modifications and answered the question required by the reviewer. I wanted to be clear that they could have done a better job to direct answer to the reviewer comments using the cover letter to do it.
Specific comments:
Line 312-313: “This suggests that the rPoIFN-β prepared in this study could be used to treat pig viral diseases.” This conclusion is too strong, and the data does not support it; The sentence should be re-written to clarify that the data was generated in an in vitro system, more in vivo tests will be needed to conclude the rPoIFN-β can be used to treat animals, such as maximum tolerated dose.
Author contributions section: “ZD carried out animal raising and blood collection”. However, the paper methodology and results section do not describe the use of blood samples in this paper. It should be corrected.
Author Response
Dear Editor,
We appreciate and accept the comments and suggestions from the two reviewers. We warmly thank the reviewer for his/her courage and appreciation of our work.
Specific comments:
1.Line 312-313: “This suggests that the rPoIFN-β prepared in this study could be used to treat pig viral diseases.” This conclusion is too strong, and the data does not support it; The sentence should be re-written to clarify that the data was generated in an in vitro system, more in vivo tests will be needed to conclude the rPoIFN-β can be used to treat animals, such as maximum tolerated dose.
Response 1. The conclusion is revised as suggested
- Author contributions section: “ZD carried out animal raising and blood collection”. However, the paper methodology and results section do not describe the use of blood samples in this paper. It should be corrected.
Response 2. The authorship file has been corrected according to the reviewer suggestions with hand signatures from each author and sent to the Editor
